# The Functioning of Na^+^-ATPases from Protozoan Parasites: Are These Pumps Targets for Antiparasitic Drugs?

**DOI:** 10.3390/cells9102225

**Published:** 2020-10-02

**Authors:** Claudia F. Dick, José Roberto Meyer-Fernandes, Adalberto Vieyra

**Affiliations:** 1Leopoldo de Meis Institute of Medical Biochemistry, Federal University of Rio de Janeiro, Rio de Janeiro 21941-902, Brazil; meyer@bioqmed.ufrj.br; 2Carlos Chagas Filho Institute of Biophysics, Federal University of Rio de Janeiro, Rio de Janeiro 21941-170, Brazil; avieyra@biof.ufrj.br; 3National Center of Structural Biology and Bioimaging/CENABIO, Federal University of Rio de Janeiro, Rio de Janeiro 21941-902, Brazil; 4Graduate Programa of Translational Biomedicine/BIOTRANS, Unigranrio University, Duque de Caxias 25071-202, Brazil

**Keywords:** ENA P-type ATPase, Type IID P-type ATPase, ATP4-type ATPase, Na^+^-ATPases, Trypanosomatids, Apicomplexa, Spiroindolones, Malaria clinical trials

## Abstract

The ENA ATPases (from *exitus natru*: the exit of sodium) belonging to the P-type ATPases are structurally very similar to the sarco/endoplasmic reticulum Ca^2+^-ATPase (SERCA); they exchange Na^+^ for H^+^ and, therefore, are also known as Na^+^-ATPases. ENA ATPases are required in alkaline milieu, as in the case for *Aspergillus*, where other transporters cannot mediate an uphill Na^+^ efflux. They are also important for salt tolerance, as described for *Arabidopsis*. During their life cycles, protozoan parasites might encounter a high pH environment, thus allowing consideration of ENA ATPases as possible targets for controlling certain severe parasitic diseases, such as Chagas’ Disease. Phylogenetic analysis has now shown that, besides the types IIA, IIB, IIC, and IID P-type ATPases, there exists a 5th subgroup of ATPases classified as ATP4-type ATPases, found in *Plasmodium falciparum* and *Toxoplasma gondii*. In malaria, for example, some drugs targeting PfATP4 destroy Na^+^ homeostasis; these drugs, which include spiroindolones, are now in clinical trials. The ENA P-type (IID P-type ATPase) and ATP4-type ATPases have no structural homologue in mammalian cells, appearing only in fungi, plants, and protozoan parasites, e.g., *Trypanosoma cruzi, Leishmania sp.*, *Toxoplasma gondii,* and *Plasmodium falciparum*. This exclusivity makes Na^+^-ATPase a potential candidate for the biologically-based design of new therapeutic interventions; for this reason, Na^+^-ATPases deserves more attention.

## 1. Introduction

In the vast majority of cases, life has evolved in high-Na^+^ environments [1], which are considered inadequate for, or injurious to, proper cellular functioning [2], notably without mechanisms that can maintain low intracellular Na^+^ concentrations in organisms surrounded by and extracellular fluid containing ~150 mM Na^+^ [3]. Thus, Na^+^ extrusion from the intracellular milieu becomes a challenge and a prerequisite for the survival, growth, and the evolution of all species. Na^+^ movement across the plasma membrane was initially described as being associated with K^+^ movements. The first Na^+^ pump described was (Na^+^ + K^+^)-ATPase. Two significant studies identified this pump: in the first, Skou [4] found that Na^+^ and K^+^ stimulated the catalysis of ATP hydrolysis by an ATPase in membrane fragments of leg nerves from the shore crab; in the second, Post & Jolly [5] demonstrated that Na^+^ extrusion and K^+^ uptake occur simultaneously in opposing directions and in an energy-dependent manner across the red-cell plasma membrane, with a stoichiometry of 3Na^+^:2K^+^.

A few years later, Proverbio et al. [6] described a ouabain-resistant Na^+^-ATPase activity that acts independently of K^+^ in isolated membranes from the outermost cortex of a guinea pig, and that was sensitive. When they studied Na^+^ fluxes through the basolateral membranes of proximal tubule cells towards the external milieu they found two mechanisms: (i) one coupled to K^+^ and sensitive to ouabain, (ii) another one not coupled to K^+^, resistant to ouabain and sensitive to ethacrynic acid [7]. More recently [8,9], the K^+^-independent and ouabain-resistant mechanism received the name of “second Na^+^ pump”, which is still accepted [10].

The finding that the diuretic, furosemide, inhibits the ATP-dependent Na^+^ efflux independently of K^+^ (named for this reason Na^+^-ATPase) with no effect on the classic (Na^+^ + K^+^)-ATPase [11] was a significant step in identifying this Na^+^ pump in several organisms. The main evolutionary advantage of a Na^+^-ATPase is its efficacy as a mechanism for Na^+^ extrusion without interference with intracellular K^+^ homeostasis. Rocafull et al. [9,12,13] purified and cloned the Na^+^-ATPase from enterocytes (ATNA); its 3D structure was proposed, and the crucial amino acids of the catalytic and Na^+^ binding sites were identified. They also demonstrated the enzyme’s sensitivity to furosemide and its resistance to ouabain. The existence of a second Na^+^ pump—besides the (Na^+^ + K^+^)-ATPase—is now widely accepted.

The Na^+^-ATPase from lower eukaryotes was first described in trypanosomatids [14,15,16]. In protozoan parasites, Na^+^-ATPase, but not the (Na^+^ + K^+^)-ATPase, was found to be essential to energize the secondary active inorganic phosphate (P_i_) transport [17,18]. In apicomplexan parasites, Na^+^-ATPase may be a therapeutic target for malaria and toxoplasmosis [19,20]. Due to the emerging importance of the Na^+^-ATPase for virulence and its potential as a therapeutic target, this review gives an overview of the recent findings regarding the role of Na^+^-ATPase in protozoan parasites.

## 2. Trypanosomatid Parasites

The first evidence of a Na^+^-ATPase in a trypanosomatid parasite was presented by Caruso-Neves et al. [14], who showed that ouabain, the specific (Na^+^ + K^+^)-ATPase inhibitor, did not completely abolish Na^+^-stimulated ATPase activity of *Trypanosoma cruzi* epimastigotes, suggesting that this parasite has a Na^+^-ATPase that is insensitive to ouabain. Iizumi et al. [15] confirmed this observation and identified the protein. The gene responsible for Na^+^-ATPase in *T. cruzi* was named *TcENA*, due to its similarity to the gene for Na^+^-ATPase from plants and fungi called ENA (ENA from *exitus natru*: the exit of sodium) [21,22]. TcENA (Figure 1) has 10 possible transmembrane domains (TMpred Server), as well as for highly conserved domains corresponding to the Type-II P-ATPases catalytic sites: (1) the signature sequence DKTGT^365^ containing aspartic acid that is phosphorylated during the catalytic cycle; (2) the domain DGFND^700^ involved in Mg^2+^ binding (green circle), and (3) the conserved TGEA^203^ sequence (red circle), as well as F^480^, K^485^ and K^504^ (orange circle), which are related to the nucleotide-binding domain. These conserved domains are related to the Type IID P-type group; therefore, their presence in *T. cruzi* allowed TcENA to be included in the Type IID P-ATPases group [20], i.e., within a family of ion-transporting ATPases that form a phosphorylated intermediate during the catalytic cycle [23,24,25]. Phylogenetic analysis allowed for the inclusion of TcENA within the unique group Type IID or ENA-type P-ATPases, which includes *Leishmania braziliensis* and *L. donovani*, *Saccharomyces cerevisiae*, and *Entamoeba histolytica* [20].

*T. cruzi*, the etiological agent of Chagas’ Disease, is a parasitic protozoan with a complex life cycle involving morphologically and functionally different stages that enable these parasites to adapt to a variety of conditions. In insect vectors, the proliferative epimastigotes form is found in the midgut, while in mammalian hosts, the gut contains the nonproliferative trypomastigote and the proliferative amastigote forms [27]. It may be that at some stages of protozoan life cycle, different parasites—not only *T. cruzi*—would have different sensitivity to drugs that target Na^+^-ATPase.

A critical role for ATPases is to contribute to the maintenance of the plasma membrane potential (ΔΨ), which is the result of asymmetrical charge distribution when concentration gradients of H^+^, Na^+^, K^+^ and Cl^−^ are established in steady state conditions. The ΔΨ of trypomastigotes forms is markedly sensitive to extracellular Na^+^ and K^+^ concentrations, and trypomastigotes have a high Na^+^-ATPase activity that contributes to a great extent in ΔΨ generation, in contrast with what happens in amastigotes. The Na^+^ gradient in amastigote forms does not influence ΔΨ [28]. These observations match *TcENA* transcription levels, which vary among the different evolutionary forms of the parasite: trypomastigote and epimastigote forms have higher *TcENA* expression than amastigote forms [15]. Moreover, the subsequent addition of varying NaCl concentrations to a suspension of epimastigotes that were initially in Na^+^-free NMG medium (140 mM *N*-methylglucamine, 5 mM glucose, 1 mM MgSO_4_, 1 mM CaCl_2_ and 10 mM Hepes-Tris at pH 7.4) resulted in immediate and pronounced depolarization [28], consistent with the observation that Na^+^ significantly stimulates ATPase activity of TcENA-expressing cells in a dose-dependent manner. Furthermore, epimastigotes overexpressing TcENA have salt tolerance, which is probably driven by their capacity for accelerated Na^+^ efflux [15].

In *T. brucei*¸ the transmembrane electrical potential (ΔΨ) also seems to be maintained by Na^+^ fluxes, even though the H^+^ gradient is principally responsible for ΔΨ generation. In the absence of ouabain, adding Na^+^ does not depolarize the membrane, indicating that the ouabain-sensitive (Na^+^ + K^+^)-ATPase has a significant role in the regulation of ΔΨ in this trypanosomatid. Successive addition of Na^+^ (5 mM) to bloodstream forms progressively depolarizes the membrane in the presence of ouabain, whereas the same conditions do not affect procyclic forms [29], indicating again that the influence of Na^+^ on ΔΨ depends on the evolutionary form. Reviewing these data, we can propose that they provide evidence for the existence of a Na^+^-ATPase together with the (Na^+^ + K^+^)-ATPase in *T. brucei*.

*T. brucei* Na^+^-ATPase was initially annotated as a putative Ca^2+^ pump (TBCA1), as was PMR2 from *S. cerevisiae*. It was soon demonstrated that *PRM2* encodes a Na^+^ pump, called ENA [30], rather than a Ca^2+^ pump. The TBCA1 protein was upregulated in the bloodstage trypomastigotes from *T. brucei*, and the cellular localization of this protein was surrounding the nucleus and the flagellar pocket, as well as the flagellum itself, in both blood stages, and to a lesser extent in culture procyclic stages [16]. Salinomycin, a monovalent cation ionophore for Na^+^, K^+^ and Rb^+^, has anti-cancer activity and an intense trypanocidal activity in *T. brucei*, once it disrupts the Na^+^ gradient, resulting in a high intracellular Na^+^ concentration that leads to cells swelling in the bloodstream forms, a condition that resembles hypo-osmotic shock. Since salinomycin did not intensify cytotoxic effects or increase intracellular Na^+^ concentration in several mammal cells lines, it has been proposed that the drug, by specifically targeting *T. brucei*, could be used as a trypanosomicide agent without causing cell damage in vertebrate hosts [31].

The presence of Na^+^-ATPase in *Leishmania* was also demonstrated 20 years ago. As with TBCA1, LdCA from *L. donovani* was initially identified as a Ca^2+^ pump because of its similarity to PMR2 from *S. cerevisiae*. Phylogenetic analysis showed that the *Leishmania* sequence ought to be included in the group of ENA ATPases rather than among Ca^+^-ATPases, as previously anticipated [16,30]. In *L. amazonensis*, ouabain completely inhibited (Na^+^ + K^+^)-ATPase without any effect on parasite proliferation. Although furosemide can inhibit the function of several transporters, the drug acts as an inhibitor of Na^+^-ATPase in *L. amazonensis* and disturbs cell proliferation, which could modify the cell volume as consequence of an increased intracellular Na^+^ concentration [32]. The potential therapeutic effect of furosemide as anti-leishmanial agent has been suggested more recently, with the advantage that furosemide has since become one of the most frequently prescribed medications globally in diuretic therapy [33].

An essential role of the Na^+^-ATPase is to maintain the Na^+^ gradient necessary to energize secondary active transport processes in trypanosomatid parasites. The furosemide-sensitive Na^+^-ATPase, but not (Na^+^ + K^+^)-ATPase, is coupled to the influx of the critical nutrient inorganic phosphate (P_i_) in *T. rangeli* and *T. cruzi* [17,18]. In both trypanosomatids, there are two P_i_ uptake mechanisms: a Na^+^-independent and a Na^+^-dependent. Regarding the last one, the Na^+^-ATPase is responsible for generating the Na^+^ gradient utilized by the Na^+^:P_i_ symporter in epimastigote forms. In *T. cruzi*, Na^+^-independent P_i_ uptake is energized by (H^+^+K^+^)-ATPase with concomitant K^+^ cycling, whereas the same P_i_ uptake mechanism is energized by an H^+^-ATPase in *T. rangeli*, demonstrating that, depending on the species, Na^+^-ATPase has selective partners to improve P_i_ uptake globally, which probably developed as a result of selective pressure.

The observation that *Leishmania* and *Trypanosoma* both have exclusive Na^+^ pumps (ENA-type Na^+^-ATPases) that are not present in mammalian cells opens up new possibilities for developing drugs against trypanosomatid parasites by exploiting biologically-based designs that do not affect the mammalian host, as proposed by Stiles et al. [16].

## 3. Apicomplexan Parasites

Infection by *Plasmodium falciparum*, the etiological agent of malaria, in the human erythrocyte induces expression of new permeability pathways (NPP) in its membrane [34]. NPP is postulated to be an anion-selective channel (also referred to as *Plasmodium* surface anion channel—PSAC), with a high permeability to a wide variety of organic and inorganic anionic molecules [35]. The physiological roles proposed for the NPP, besides uptake of essential nutrients, include the efflux and influx of solutes from the parasitized erythrocyte. For example, the flux of Na^+^ and K^+^ via the NPP induced by *Plasmodium* infection substantially increases the normally weak Na^+^ and K^+^ permeability of the membrane, increasing [Na^+^] and decreasing [K^+^] in the cytosol [36]. *Plasmodium* infection causes a profound perturbation of ion homeostasis in the host cell, due to the induction of NPP, leading to Na^+^ and K^+^ redistribution across the membrane, whereas it has little effect on Cl^−^, pH and Ca^2+^ [37]. Parasite growth is not affected by the erythrocyte Na^+^ concentrations, possibly due to the existence of efficient mechanisms responsible for the extrusion of intracellular Na^+^ from the protozoa, likely the Na^+^-ATPase—in exchange for H^+^—proposed by Kirk [37], despite osmotically induced morphological alterations [38]. The inwardly directed Na^+^ gradient could energize uphill Na^+^-dependent transporters by the parasite [39]. P_i_ uptake by the malaria parasite is electrogenic, involving an influx of positive charge that uses the Na^+^ gradient across its plasma membrane. This Na^+^:P_i_ symporter has a stoichiometry of 2Na^+^:1H_2_PO_4_ [40]. Also, in *Plasmodium*, the protein PfATP4 was initially characterized as a member of a new subfamily of P-type Ca^+^-ATPases [41]. It was later demonstrated that PfATP4 possesses sequence similarities to Na^+^-extruding ENA-type ATPase [19], and is localized on the parasite plasma membrane, therefore being suggested as the protein responsible for Na^+^ efflux [42].

As discussed above regarding trypanosomatids, Na^+^-ATPases from apicomplexan parasites (ATP4 P-type ATPases) have no functional homologs to animal cell P-ATPases, and thus present attractive potential targets for drugs [20,25,37]. Inhibition of ATPases can in turn inhibit pathogen growth; specific drugs against P-type ATPase could then be designed [25]. However, parasite growth in cultures in the presence of these compounds for a prolonged time results in the parasites developing a degree of resistance associated in each case with PfATP4 mutations [43]. In this group of compounds, synthetic compounds related to the spiroindolone class have a favorable pharmacological profile. The so-called spiroindolones are antimalarial drug candidates because they can specifically inhibit the P-type ATPase PfATP4 activity [42,43,44], and clinical trials are under way.

Spiroindolones increase intracellular Na^+^ concentrations ([Na^+^]_in_) in a dose-dependent manner, and the disruption of Na^+^ regulation occurs in conjunction with inhibition of parasite proliferation. Moreover, mutations in PfATP4 induced by spiroindolones confer resistance to inhibition of membrane-associated ATPase activity promoted by the drugs, and increases [Na^+^]_in_ [44]. However, some drug resistance has a high fitness cost [45] because it provokes basal intracellular Na^+^ doubles in these mutants. Spiroindolones provide several benefits when used as drugs: (i) they present a rapid clearance *in* vivo, (ii) they act exclusively on infected erythrocytes, therefore being selective, and, (iii) parasites that develop resistance to this drug also suffer the above-mentioned high fitness cost, once a high [Na^+^]_i_ causes cellular stress. It is interesting that some PfATP4 mutations are not associated with growth defects [46]; thus, the mutation-induced alterations in basal [Na^+^]_in_ would be different depending on which amino acid has changed.

Cipargamin (also known as KAE609 or NITD609) is a spiroindolone that is very useful in malaria-infected humans [47]. As pointed out by Zhang et al. [48] KAE609 can directly kill the parasite by inhibiting key metabolic pathways or through oxidative damage and, additionally, by disrupting the structure of infected erythrocytes. These mechanisms together promote a remarkable rapid clearance of parasites [48]. The Na^+^-dependent ATPase activity of membrane fractions of *Plasmodium* is also inhibited by cipargamin, an effect that is less pronounced in cells with a single mutation in PfATP4, demonstrating a direct effect of cipargamin on Na^+^-ATPase activity [49]. Spiroindolones cause swelling of malaria parasites in a Na^+^-dependent way, which may be due to an increase in intracellular Na^+^. After inhibition of PfATP4 the Na^+^ ions that leak into the cell can no longer be pumped out, leading to an increase in intracellular Na^+^ and cell swelling [50]. Another class of drugs, the dihydroisoquinolones (DHIQs), are also effective inhibitors of PfATP4. One (+)-enantiomer (SJ733) binds to PfATP4 as a target, and treatment with (+)-SJ733 induces eryptosis or senescence only of erythrocytes infected with *Plasmodium*, leading to a rapid clearance of the parasite in vivo. As seen with related compounds, strains resistant to (+)-SJ733 possess mutation of the PfATP4 gene, and again these parasites carry a high fitness cost for survival [45].

Besides spiroindolones and dihydroisoquinolone (DHIQ), another class of antimalarial drugs, the pyrazolemides [51], share a standard mode of action with the ones previously mentioned. All these drugs induce a rapid influx of Na^+^ in trophozoite stages of *P. falciparum* as a consequence of PfATP4 inhibition. Das et al. [52] demonstrated that an increase in [Na^+^]_in_ inhibits a putative cholesterol pump, impairing the morphology of the parasite. In this way, PfATP4 inhibition by either spiroindolones or pyrazoleamides leads to premature disruption of a finely tuned process, resulting in an influx of Na^+^ into parasite cytoplasm, premature inhibition of cholesterol pump and induction of schizogony that would result in parasite death. Cipargamin and (+)-SJ733 are now potential drugs for treating malaria, and both are in clinical trials [53] (Table 1 for ongoing cipargamin studies).

Regarding another apicomplexan, *Toxoplasma gondii*, the agent causing toxoplasmosis, the spiroindolone cipargamin has proved to be effective in disrupting parasite proliferation in vivo and in vitro [61]. However, the characterization of a homologue of PfATP4, the so-called TgATP4, has only recently been published, and parasites disrupted by TgATP4 remain viable and can cause the disease in vivo with reduced virulence [20].

## 4. Other Protozoan Parasites

In *Giardia intestinalis*, the etiological agent of giardiasis, ΔΨ is dependent on H^+^/Na^+^/K^+^ gradients [62]. Na^+^ also has an essential role in the maintenance of a steady-state intracellular pH (pH_in_). This effect of Na^+^ on pH_in_ was suggested to be maintained by a Na^+^/H^+^ exchanger sensitive to the inhibitor, amiloride [63], but as it is not entirely abolished, one cannot exclude the effect of an ATP-driven Na^+^ pump on maintenance of the Na^+^ gradient. Regarding another pathogen protozoan, *Entamoeba histolytica*, the presence of a Na^+^ pumping mechanism in its plasma membrane was suggested by Bakker-Grunwald et al. [64] because they could show that amiloride has a profound effect on Na^+^ homeostasis [65] without affecting intracellular K^+^. However, the nature of the pump is still not understood. In another amoeba, *Acanthamoeba castellanii*, the Na^+^/H^+^ exchanger blocker, cariporide, partially inhibits excystation, indicating that Na^+^ fluxes have an essential role in pH regulation and trophozoite formation [66]. In silico analysis suggests that these parasites could have an Na^+^-ATPase (Figure 2) related to PfATP4, and as in the case of *Plasmodium*, potential Na^+^-ATPase inhibitors could treat these protozoan diseases.

## 5. Conclusions

Described initially as Ca^2+^-ATPases due to their similarity to Ca^2+^-ATPases of the endoplasmic reticulum (SERCA), the ENA P-type ATPases and ATP4 P-type ATPases of protozoan parasites are responsible for maintaining low cytosolic Na^+^. These pumps are absent in mammals, which makes them a good drug target. In trypanosomatids, Na^+^ fluxes maintained by the ENA-type pump seem to be related to ΔΨ maintenance and to energizing secondary active transport, such as P_i_ transporter (Figure 3). Besides, inhibition of Na^+^-ATPase with furosemide disrupts cell proliferation in vitro, demonstrating that this pump deserves more attention if it can be exploited in the biologically-based design of new therapeutic interventions for parasite diseases. In Apicomplexa, inhibition of Na^+^-ATPase is well studied, with drugs targeting PfATP4 such as spiroindolones, causing a disruption on Na^+^ homeostasis. In this way, these drugs could be used in the future for treating malaria. Table 1 presents ongoing clinical trials—described in details in [53]—in which subjects (healthy or infected with uncomplicated malaria) were, or still are, treated with cipargamin, PfATP4 being the target of the drug.

## Figures and Tables

**Figure 1 cells-09-02225-f001:**
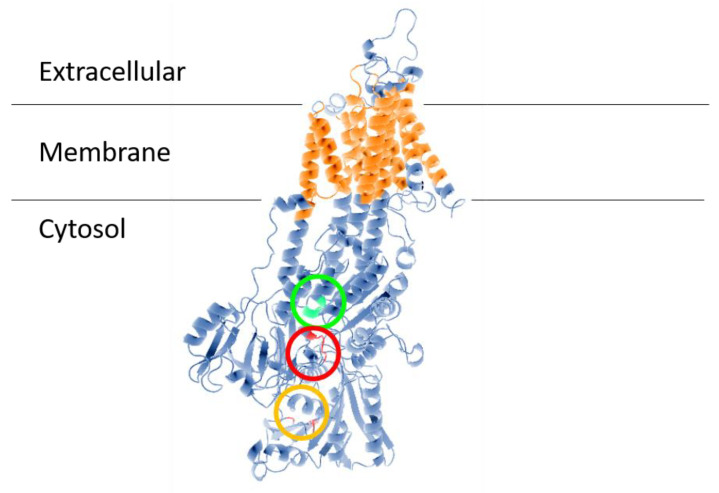
Structural model of ENA ATPase from *T. cruzi* (TcENA). The model was constructed using the protein structure prediction PHYRE (www.sbg.bio.ic.ac.uk/phyre/) [26], based on the model of the SERCA ATPase and visualized with the standard molecular viewer PyMOL 2002 (DeLano, W.L. The PyMOL Molecular Graphics System, DeLano Scientific, San Carlos, CA, USA; http://pymol.sourceforge.net/). The 10 transmembrane helices are shown in orange, and the key residues mentioned in the text are highlighted by a red circle (conserved TGEA domain), an orange circle (F^480^, K^485^, K^504^ residues related to nucleotide binding); and a green circle (DGFND^700^ domain involved in Mg^2+^ binding).

**Figure 2 cells-09-02225-f002:**
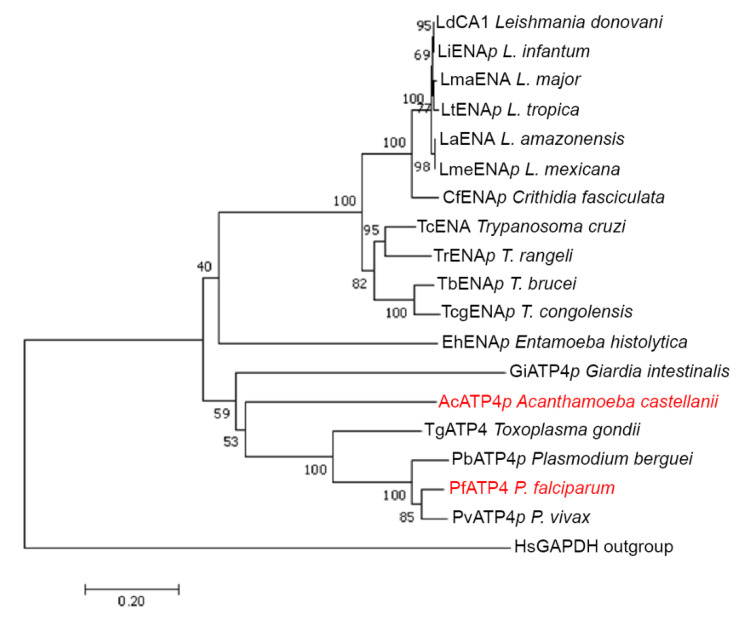
Phylogenetic analysis of Na^+^-ATPases from protozoan parasites. Amino acid sequences from different unicellular eukaryotic species were aligned, and phylogenetically analyzed with MEGA 5.2.2 software. The ENA ATPase members (with their respective GenBank information) are as follows: LdCA1 from *L. donovani* (GenBank: LdCL_350026000), LiENA putative (p) from *L. infantum* (GenBank: LINF_350025900), LmaENA from *L. major* (GenBank: LmjF.35.2080), LtENA p from *L. tropica* (GenBank: LTRL590_350026000), LaENA from *L. amazonensis* (GenBank: LAMA_000787300), LmeENA p from *L. mexicana* (GenBank: LmxM.34.2080), CfENA p from *Crithidia fasciculata* (GenBank: CFAC1_300065600), TcENA from *T. cruzi* (GenBank: Q76DT8_TRYCR), TrENA p from *T. rangeli* (GenBank: TRSC58_02255), TbENA p from *T. brucei* (GenBank: Tb927.9.15460), TcongENA p from *T. congolensis* (GenBank: TcIL3000.A.H_000710900), EhENA p from *E. histolytica* (GenBank: EHI5A_015420), GiENA p from *G. intestinalis* (GenBank: ESU43140.1), AcENA p from *A. castellani* (GenBank: ACA1_313610), TgATP4 from *T. gondii* (GenBank: TGME49_278660), PbATP4 p from *P. berguei* (GenBank: PBANKA_0610400), PfATP4 from *P. falciparum* (GenBank: Q9U445_PLAFA), PvATP4 p from *P. vivax* (GenBank: PVP01_1311100). The phylogenetic analysis was performed using the human GAPDH (GenBank: NP_002037.2) as outgroup sequence, due to its distance from the Type II P-type ATPase proteins.

**Figure 3 cells-09-02225-f003:**
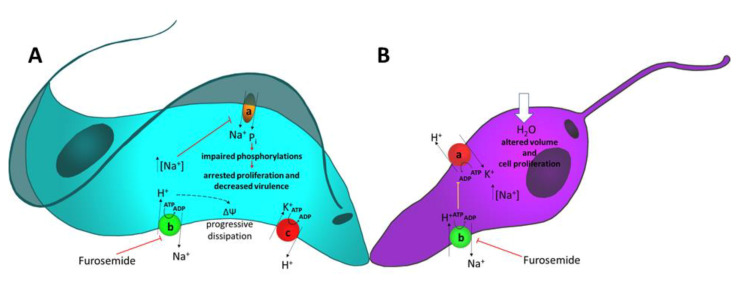
Proposed model for intracellular events in *T. cruzi* trypomastigotes form (**A**) and *L. amazonensis* procyclic forms (**B**), triggered by the inhibition of the ENA-type ATPases. Furosemide is used as a model for drugs targeting the pump in protozoan parasites without affecting human host cells. In A, inhibition of Na^+^-ATPase (red hammer) increases [Na^+^]_in_ and decreases the uptake of P_i_ by the Na^+^:P_i_ symporter [18], thus impairing metabolic phosphorylations, arresting proliferation and decreasing virulence. Additionally, modification of the transmembrane Na^+^ gradient (a higher intracellular Na^+^) leads to progressive dissipation of the ΔΨ [28] (dashed arrow). In B, inhibition of Na^+^-ATPase (red hammer), increases [Na^+^]_in_ and thereby the influx of water [67], with subsequent disruption of normal cell volume and impaired proliferation [32]. The decrease in H^+^ influx as the result of inhibition of Na^+^-ATPase decreases turnover of the plasma (H^+^+K^+^)-ATPase [68]. Progressive decrease in K^+^ influx would contribute to arresting growth and cell proliferation.

**Table 1 cells-09-02225-t001:** Clinical trials on cipargamin.

Author	Phase	Target	Population	Healthy/Infected	Reference
NJ White	2	PfATP4 Na^+^-ATPase	Adults	Uncomplicated *P. vivax* or *P. falciparum* malaria	[47]
FJ Leong	1	PfATP4 Na^+^-ATPase	Male adults	Healthy	[54]
DS Stein	1		Male adults	Healthy	[55]
SW Huskey	1		Adults	Healthy	[56]
TT Hien	2a		Male adults	*P. falciparum* malaria	[57]
Study CKAE609X2202	2		Adults	*P. falciparum* malaria	[58]
Study CKAE609A2109	1		Adults	Human challenged model-induced *P. falciparum* malaria in healthy adults	[59]
MF Chuglay CNCT0025L3086	2		Adults		[60]
SAM Bouwman	The authors searched for preclinical studies and clinical trials using cipargamin				[53]

Modified from ref. [53] with permission.

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
