# Peer review of "The Functioning of Na+-ATPases from Protozoan Parasites: Are These Pumps Targets for Antiparasitic Drugs?"

_cells, 2020, doi:10.3390/cells9102225_

Round 1
Reviewer 1 Report
This is an informative- overall well-presented and organized review of the current knowledge of structural and functional characteristics of ENA ATPases, with a focus on their merit as prospective drug targets in protozoan (antiparasitic) pathogenesis.
The review is organized in 5 sections, 1. Introduction, 2. Trypanosomatid parasites, 3. Apicomplexa parasites, 4. other protozoan parasites, and 5. Conclusion. Addressing the following points should help increase clarity and impact.
- The novelty of the most recent findings covered by the review could be better emphasized. Recent/ongoing clinical trials/pharmacological advances and/or proposed new target domains could be summarized in table/figures.
- Section 2 reviews structural features of ENA ATPases that do not appear to be limited to trypanosomatids (figure 1). Please clarify/reorganize if needed.
- For the non-expert reader, a brief descriptive of stages (trypomastigotes, epimastigotes….) would be helpful.
- Thorough proofing is recommended, including abstract
- Title should be revised. “seeking for a function” does not adequately describes the content/message of this review.
- Adding identity of organism next to sequence labels in Figure 2 would improve readability. Highlight (possibly with color) the sequence that are mentioned/compared in the text (e.g. PfATP4 and AcENA, ….) so that the reader can easily assess how closely related they are. Explain the choice of human GAPDH as the outgroup.
- Please clarify “Figure (1, 2, or 3) Dick et al., 2020”. Have those figures been previously published?
- Increase font size and contrast of labels in figure 3
- “conclusion” should be numbered 5 (rather than 4).
- Minor formatting inconsistency in the references.
- Line 92. Define “NMG”
- Line 57 “the review aims gives”
Reviewer 2 Report
This review deals with an interesting topic but unfortunately contains a number of inaccuracies, including the designation of PfATP4 and TgATP4 as ENAs. Specific comments are listed below.
The title is confusing. Is it ENAs that are being referred to as ‘second’ pumps? Is the Na+/K+ ATPase the ‘first’? Many parasites will not encode a Na+/K+ ATPase at all. In P. falciparum, PfATP4 (not an ENA) is likely to be the only ATP-driven Na+ transporter on the parasite plasma membrane.
Statements such as “ENA ATPase is required at high external pH values, where other antiporters cannot mediate an uphill Na+ efflux”, with no organism specified, are overgeneralisations and should be replaced with specific, accurate statements.
Line 50: “ENA from enterocytes”? Earlier the manuscript states that ENAs are “exclusive to fungi, plants, and protozoan parasites”.
The word ‘pump’ typically refers to an ATP-driven transporter, and not to secondary active transporters such as the Na+/H+ exchanger (NHE) or Na+/K+/Cl- symporter. The authors should define the word ‘pump’ and use it accurately throughout the manuscript.
Line 121: “adding furosemide…disturbed cell proliferation in vitro, demonstrating a crucial role of the furosemide-sensitive Na+ pump in cell proliferation”. Furosemide has been reported to inhibit quite a number of membrane transport proteins – the authors should be more cautious in their interpretation.
A detailed phylogenetic analysis published recently did not support the designation of PfATP4 and TgATP4 as ENAs. Rather, these proteins belong to a distinct subgroup of type II P-type ATPases (PMID: 30723156).
The sentence beginning on line 169 does not make sense and needs revising.
Line 178: Ref. 40 is not an appropriate reference for the fitness cost of PfATP4 mutant parasites.
Regarding fitness costs of PfATP4 mutant parasites, numerous mutant parasites have been generated by multiple groups and some 40 different mutations in PfATP4 have been reported. Only a small number of mutant parasites have been studied in fitness experiments. Some PfATP4 mutant parasites have been reported not to display a growth defect (see Flannery et al. ACS Chem Biol). It is likely that some mutations will be associated with growth defects and that some will not. Likewise, there will be differences in the degree to which different mutations affect the parasite’s resting internal Na+ concentration.
Cipargamin is believed to kill parasites via inhibition of PfATP4. I do not understand the sentence that starts on line 184 or the one after it. What ‘dual mechanism’ are the authors referring to?
The authors refer to a recent publication on TgATP4 but have misinterpreted it. T. gondii parasites remain viable when they do not express TgATP4 and display only a minor reduction in virulence. It is misleading to suggest that cipargamin kills T. gondii parasites via inhibition of TgATP4 (line 210), that TgATP4 has a crucial role in T. gondii survival (line 210) or that PfATP4 inhibitors could be used to treat T. gondii infections (line 254).
Round 2
Reviewer 2 Report
The authors have made a number of improvements to the manuscript.
However, there are still some issues with accuracy, clarity and flow.
1) In the abstract - I am not sure that it makes sense to refer to structural similarity/structural homologues when no high-resolution structures exist for the relevant proteins.
2) Line 81 onwards - the 'second Na+ pump' being described sounds like the ethacrynic acid sensitive Na+/K+/Cl- symporter, which is not an ATPase and not a pump. To avoid confusion about what this protein is, could the authors please state its identity. If it is the Na+/K+/Cl- symporter, it should not be referred to as a pump or an ATPase.
3) Line 241 - the authors state that a cation ionophore inhibits Na+ flux, which does not make sense. Presumably the cation ionophore increases the flux of Na+ down its electrochemical gradient to such an extent that the Na+-extruding ATPase cannot effectively counter the influx and can no longer maintain a low intracellular Na+ concentration.
4) Line 266 - Pi flux is not energised by an ATPase. It is energised by the ion gradient maintained by the ATPase.
5) Lines 269-270 - the meaning is not clear.
6) The section on the evidence that spiroindolones inhibit PfATP4 and the effects that they have requires further revision.
-For example, the authors state that "spiroindolones...specifically inhibit the P-type ATPase PfATP4" (suggesting that PfATP4 is the confirmed target), and then in the next sentence talk about something "raising the possibility" that PfATP4 is the target.
-Cipargamin (rather than spiroindolones in general) should be introduced as the clinical candidate before the Table appears.
-Line 391 - I still do not understand what metabolic pathway is being referred to here, and what evidence there is for cipargamin causing oxidative damage.
-Line 397 - "may be due to an increase in Na+ influx following inhibition of PfATP4". This is an unlikely scenario - there is no reason to believe that the magnitude of the inward Na+ 'leak' has changed. Na+ ions will always be leaking into cells down their electrochemical gradient (e.g. via Na+-coupled transporters). When PfATP4 is inhibited, the Na+ ions that leak into the cell can no longer be pumped out again, resulting in an increase in the intracellular Na+ concentration.
7) If the authors agree with the new evidence that PfATP4 and TgATP4 are ATP4-type ATPases and not ENAs, why do these proteins appear in a Figure labelled "Phylogenetic analysis of ENA ATPases..."?
